# Examining pulmonary TB patient management and healthcare workers exposures in two public tertiary care hospitals, Bangladesh

**Md. Saiful Islam**[1,2]*, **Sayera Banu**[1], **Sayeeda Tarannum**[1], **Kamal Ibne Amin Chowdhury**[1], **Arifa Nazneen**[1], **Mohammad Tauhidul Islam**[1], **S. M. Zafor Shafique**[1], **S. M. Hasibul Islam**[1], **Abrar Ahmad Chughtai**[2], **Holly Seale**[2]

**1** Emerging Infections Program, Infectious Diseases Division, icddr,b, Dhaka, Bangladesh, **2** School of Population Health, Faculty of Medicine and Health, University of New South Wales, Sydney, Australia

* saiful@icddrb.org

**Data Availability Statement:** Data are available from the data repository committee at icddr,b. The dataset underlying the findings described in the

## Abstract

Implementation of tuberculosis (TB) infection prevention and control (IPC) guidelines in public tertiary care general hospitals remain challenging due to limited evidence of pulmonary TB (PTB) patients' duration of hospital stay and management. To fill this evidence gap, this study examined adult PTB patient management, healthcare workers' (HCWs) exposures and IPC practices in two public tertiary care hospitals in Bangladesh. Between December 2017 and September 2019, a multidisciplinary team conducted structured observations, a hospital record review, and in-depth interviews with hospital staff from four adult medicine wards. Over 20 months, we identified 1,200 presumptive TB patients through the hospital record review, of whom 263 were confirmed PTB patients who stayed in the hospital, a median of 4.7 days without TB treatment and possibly contaminated the inpatients wards. Over 141 observation hours, we found a median of 3.35 occupants present per 10 m$^2$ of floor space and recorded a total of 17,085 coughs and 316 sneezes: a median of 3.9 coughs or sneezes per 10 m$^2$ per hour per ward. Only 8.4% of coughs and 21% of sneezes were covered by cloths, paper, tissues, or by hand. The HCWs reportedly could not isolate the TB patients due to limited resources and space and could not provide them with a mask. Further, patients and HCWs did not wear any respirators. The study identified that most TB patients stayed in the hospitals untreated for some duration of time. These PTB patients frequently coughed and sneezed without any facial protection that potentially contaminated the ward environment and put everyone, including the HCWs, at risk of TB infection. Interventions that target TB patients screening on admission, isolation of presumptive TB patients, respiratory hygiene, and HCWs' use of personal protective equipment need to be enhanced and evaluated for acceptability, practicality and scale-up.

paper cannot be shared publicly due to ethical restrictions related to protecting study participants' privacy and icddr,b's data access policy (https://www.icddrb.org/policies). icddr,b has a data repository maintains by the research administration. A copy of the complete dataset (anonymized and decoded) of this study will remain at the data repository. Interested researchers may contact Ms. Armana Ahmed, head of research administration (aahmed@icddrb.org), for approval and data access.

**Funding:** This research protocol was funded by the United States Centre for Disease Control and Prevention (CDC), through the cooperative agreement grant number 5U01GH1207. MSI and SB received this award. The funders had no role in study design, data collection and analysis, decision to publish, or preparation of the manuscript.

**Competing interests:** All authors report no conflicts of interest relevant to this article.

## Introduction

Public tertiary care hospitals in low-and middle-income countries (LMIC) are generally densely occupied, raising the possibility of infectious and susceptible individuals being co-located in the same area. This creates a unique challenge when reducing the risk of nosocomial transmission of tuberculosis (TB) [1, 2]. Due to limited implementation of TB infection prevention and control (IPC) healthcare measures in some LMIC settings, healthcare facilities are now a common infection source with drug-sensitive and drug-resistant strains of TB [3–5]. Factors that influence the risk of nosocomial TB infections include the presence of untreated pulmonary TB (PTB) patients, frequency of patients with infectious TB not covering their coughs, inadequate room ventilation, respiratory hygiene, and the inadequacy of TB IPC measures [6, 7].

These PTB patients often remain untreated due to a low index of suspicion and a lack of proper screening and diagnostic evaluation [8]. A lack of screening procedures for patients with presumptive TB on admission may lead to a delayed diagnosis that allows the opportunity for nosocomial TB transmission from untreated PTB patients to other hospital wards' occupants.

The international TB IPC guidelines recommend that healthcare settings in LMIC implement the following control measures: i) triaging of symptomatic TB patients or patients with PTB disease; ii) separation or isolation of patients with infectious TB; iii) rapid diagnosis and initiation of treatment; iv) supporting adherence with respiratory hygiene measures and v) ensuring adequate ventilation [9]. Successful implementation of these administrative and personal level IPC measures, including personal protective equipment (PPE), can reduce nosocomial transmission of TB [10].

In Bangladesh, tertiary care general hospitals admit presumptive TB patients for diagnosis and occasionally provide treatment to TB patients with comorbidities. When this study was conducted, there were no or limited studies about PTB patient management and healthcare workers' (HCWs) exposures to untreated TB patients in public tertiary care general hospitals, Bangladesh. Due to the lack of evidence about PTB patients' stay in Bangladesh's public tertiary care general hospitals, the implementation of TB IPC measures in these facilities remained challenging [11, 12]. This study aimed to assess PTB patients' length of hospital stay and management and healthcare workers' exposures to PTB patients, and the use of PPE in two public tertiary care hospitals in Bangladesh.

## Methods

### Study design, study site, and data collection

A mixed-method study was conducted between December 2017 and September 2019, which included structured observations, hospital record reviews, and in-depth interviews with HCWs. The use of these multi-method data collection tools allowed us to triangulate and cross-check the data that emerged from various sources [13]. This mixed-method approach also allowed to compare quantitative and qualitative data that increased the rigor of the research, which was less likely to be achieved using a single data collection tool [14]. A team of six social scientists, epidemiologists and a medical technician with several years of IPC research experience collected data in two public tertiary care teaching hospitals (Hospital A and Hospital B) in Bangladesh. Hospital A was a 1,200-bed tertiary level hospital, the second-largest hospital in the country that served more than 2,000 patients per day [15]. Hospital B was a 1000-bed tertiary level hospital, and around 1,650 patients were admitted per day in the hospital [16]. The rationale for selecting these hospitals was based on the level of care provided

(tertiary), that they were known to provide diagnosis and care to TB patients, and convenience (there was already an established relationship and research occurring within these facilities) [17, 18]. Moreover, these two hospitals were the largest tertiary care public hospitals that serve around 27 million people from 14 districts in Rajshahi and Barishal divisions in Bangladesh [19–22].

**Observations.**   After seeking written approval from the hospital directors and unit heads, the team conducted observation in adult general medicine wards (two male medicine wards and two female medicine wards). The aim of observation was to systematically document patients and HCWs' behaviors and practices occurring in a naturalistic environment [23]. We conducted 6 to 8 hours of unscheduled observation per day for six days in study wards. To capture the variety of exposures at different times of a shift, the team conducted one to three hours of observations in the morning, in the evening and at night (total: 69 sessions:17 to 18 sessions per ward). Based on our prior experience working in these hospitals, we anticipated the number of occupants and IPC practices may vary between male medicine wards and female medicine wards and therefore, we documented the data separately. Through structured direct observation, the team recorded information on the number of HCWs present, duration of PPE use, number of HCWs wearing a mask or N95 respirator, and the number of presumptive TB patients identified. We recorded information related to patient screening for cough, the availability of masks for patients, the type of masks used, the availability of separate waiting areas for patients with suspected TB, and the triaging processes for patients with suspected TB in the outpatient departments.

**Hospital record review.**   After seeking written approval from the hospital directors and unit heads, we reviewed hospital records from December 2017 to July 2019. Two team members with several years of experience in hospital surveillance were deployed in the study hospitals. Each day, the team members visited the adult medicine wards, reviewed the patient registries, and documented the number of presumptive TB patients on admission, number of sputa collected from suspected TB patients, number of suspected TB patients sent for a chest x-ray (CXR) (including the percentage with an abnormal CXR), number of patients diagnosed with TB among those suspected, number of suspected TB patients with at least one positive smear, and number of suspected TB patients sent to GeneXpert testing and the percentage testing positive. If there was any information missing in the patient file, the team talked to the patient's family caregivers and collected the information.

**In-depth interviews.**   Seventeen in-depth interviews were undertaken with purposefully selected senior and junior level HCWs from the medicine wards, labs and directly observed therapy, short-course (DOTS) clinics. Four field team members with several years of experience in qualitative research conducted the interviews to gain an understanding of the existing PTB management practices along with what precautions they took to prevent TB transmission in the hospitals. All but one interviews were recorded and transcribed. For the participant that refused to be recorded, and the team took detailed hand notes. The interviews aimed to capture system factors including existing practices of PTB case finding, patient admission practices, how space is allocated, and the factors that are considered while allocating space for suspected/confirmed PTB patients and discharge, along with what precautions they took to prevent TB transmission.

## Analysis

We descriptively analyzed the observation and the hospital record review data using the open-source statistical package R version 3.6.3 (R Foundation for statistical computing, Vienna, Austria, Available at https://www.R-project.org/). For continuous variables such as time spent

in the TB ward, waiting time of the TB patients in the DOTS clinic, we added the time spent for each activity and divided it by the total time by the number of times the activity was observed. Results are presented by median duration of the activity [24]. Extrapulmonary TB patients were excluded from the analysis because of their limited role in TB transmission in hospital settings.

We used logistic regression to estimate odds ratios with 95% confidence intervals (CIs) to determine factors associated with the higher length of hospital stay. The PTB patients' median length of hospital stay was 4.7 days. To form a dichotomous outcome, patients who stayed 4.7 days or less in the hospital was coded as '0' and patients who stayed more than 4.7 days was coded as '1' and compared. Factors with a p-value <0.20 in the binary logistic regression analyses were included in a multiple logistic regression model to adjust for confounders.

For the qualitative data, the team read the transcribed data and the field notes line by line several times and developed a code list. The lead author entered the transcribed interviews into text organizing software NVivo and coded them according to the code list. The coded data were then summarized, translated into English and categorized under predefined and emerging themes [25]. The pre-coded themes and sub-themes were based on study objectives and interview guidelines.

## Ethics

We sought informed written consent before the interviews. We also sought informed written consent from hospital directors and unit heads to review the medical records. The behaviours and practices we observed in the hospital wards were considered public behaviours, and the ethics committee waived the need for informed consent from the participants in this case. The institutional review board of the International Centre for Diarrheal Diseases Research, Bangladesh (icddr,b), and the University of New South Wales ethical review committee approved the study (PR-16090 and HC # HC180517).

## Results

### Observation

The team conducted 141 hours of structured observation across 69 sessions: 69 hours in two adult male medicine wards (35 sessions) and 72 hours in two adult female medicine wards (34 sessions). There were three confirmed and seven presumptive PTB patients present during the observation in both male and female medicine wards. Table 1 includes the study ward characteristics and occupancy. During the observation periods, there was a median of 86 (IQR: 164.75) occupants per session per ward: 24 (IQR: 52) patients, 46.5(IQR: 109.25) family caregivers including children, and 10 (IQR:10) HCWs (Fig 1). Fig 1 also shows the number of family caregivers were more than twice the number of patients. A median of 3.35(IQR: 2.87) occupants were present per 10 m$^2$ of floor space, with the highest number of occupants observed in female medicine wards in Hospital A; 7.0 (IQR: 0.71) occupants per 10 m$^2$ of floor space (Table 1).

All the study wards were naturally ventilated and had open floor plans. There were no separate or isolation rooms for presumptive or confirmed PTB patients in the hospitals. The team recorded a median of 15 people (IQR: 15) coughing during the observation period, most of whom were patients (median 8, IQR: 10) and their caregivers (median 6, IQR: 8). The team recorded a total of 17,085 coughs and 316 sneezes during the observation, a median of 3.9 (IQR: 4.59) coughs or sneezes per 10 m$^2$ per hour per ward. Only 8.4% of (1,437/17,085) coughs, and 21% (66/316) of sneezes were covered by cloths, paper, tissues, or by hand.

**Table 1.  Characteristics, ward occupancy and ventilation of four wards in two public tertiary care hospitals, 2019.**

| Characteristics | Hospital A | | Hospital B | |
|---|---|---|---|---|
| Ward structure | MMW | FMW | MMW | FMW |
| Ward area for patients | 410.26m$^2$ | 507.52m$^2$ | 193.75m$^2$ | 212.5m$^2$ |
| Volume of airspace | 1452.31m$^3$ | 1811.85 m$^3$ | 680.05m$^3$ | 733.1m$^3$ |
| Median number occupants per 10 m$^2$ | 3.0 (IQR:2.05) | 7.0 (IQR:0.71) | 3.2 (IQR:1.64) | 2.5 (IQR:0.89) |
| Median number of HCWs per ward | 8.0 (IQR:7.5) | 19.5 (IQR:19) | 10 (IQR: 8.25) | 9.0 (IQR:8) |
| Median number of family caregivers and children | 72 (IQR:44) | 219 (IQR:24) | 31.5 (IQR: 24.3) | 30 (IQR:16) |
| Median number of patients | 26 (IQR:37.5) | 105 (8.75) | 22 (IQR:9) | 16 (IQR:4) |
| Number of beds | 38 | 70 | 27 | 20 |
| Number of doors | 3 | 4 | 4 | 3 |
| Median number of doors closed | 2 (IQR0.0) | 1.5 (IQR:1.0) | 2.5 (IQR:1.0) | 1 (IQR:1.0) |
| Number of windows | 21 | 27 | 16 | 11 |
| Median number of windows closed | 13.5 (IQR: 2.0) | 14 (IQR: 6.25) | 8 (IQR: 2.75) | 10 (IQR:2.0) |
| Number of ceiling fans | 24 | 34 | 20 | 13 |
| Type of ventilation | Natural | Natural | Natural | Natural |
| Air change per hour | 5.4 | 14.4 | 19.2 | 9 |
| Number of coughing persons | 20 (IQR:9) | 49 (IQR: 21.25) | 14 (IQR:5.5) | 8 (IQR:7) |

MMW-Male medicine ward, FMW-Female medicine ward, IQR-Interquartile range

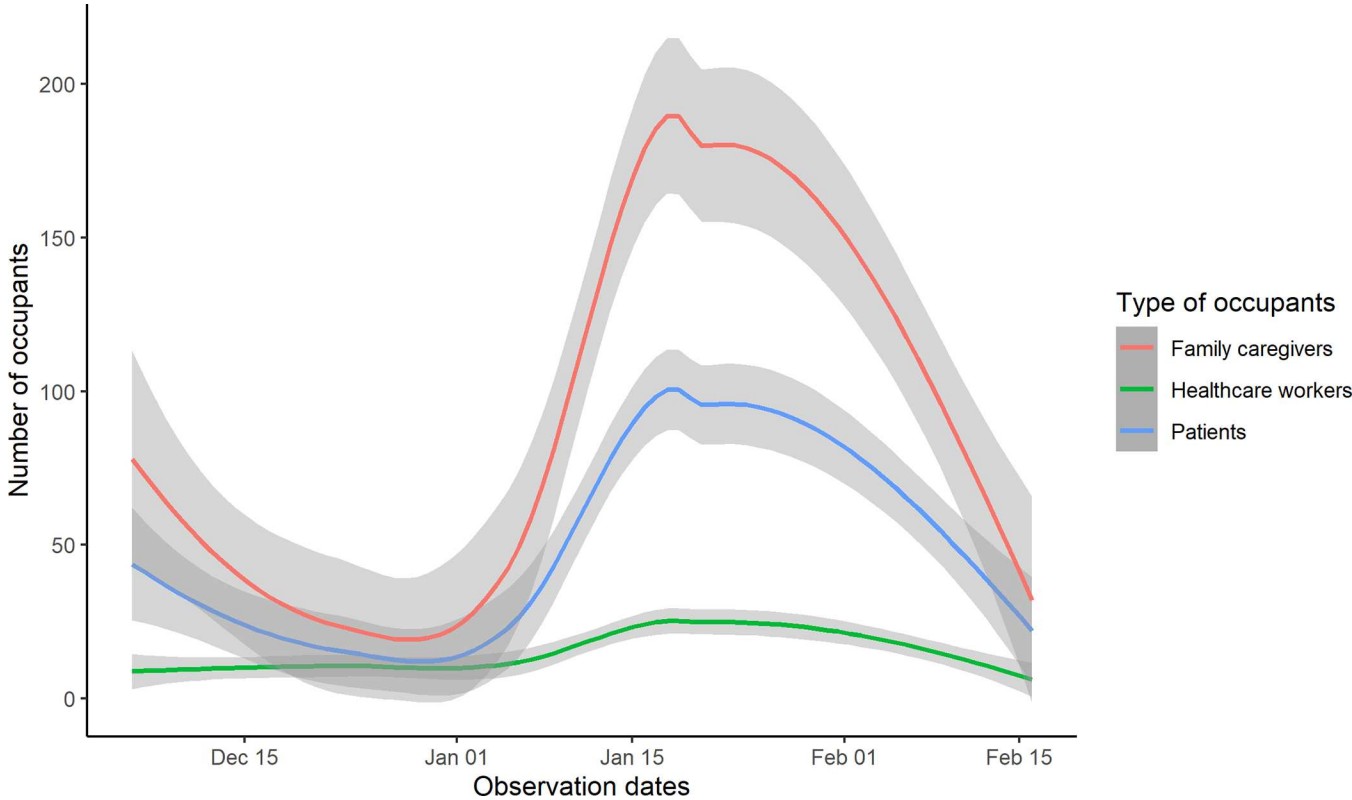

**Fig 1.  Distribution of patients, attendants and HCWs presence in medicine wards in two hospitals, Bangladesh.**

A median of three doctors (IQR: 5), four nurses (IQR: 5) and three ancillary workers (IQR: 2) were exposed to the air space per observation session. Each doctor spent a median of 46 minutes (IQR: 227), nurses spent 149 minutes (IQR: 193), and ancillary workers spent 46 minutes (IQR: 74) per observation session. None of the HCWs was observed to wear N95 respirators, and only three wore cloth masks. None of the patients was observed to put on a cloth or surgical mask.

## Hospital record review

There were 1,200 patients identified as having presumptive TB in the study wards over 20 months of which 22% (263/1200) were diagnosed with confirmed PTB. The median age of the patients was 52 (IQR: 30), and 55% (664/1200) were male. Among the presumptive TB patients, 96% (1151/1200) were recommended for TB sputum test, 82% (982/1200) for CXR, and 52% (624/1200) for GeneXpert test. Seventy eight percent (205/263) of the confirmed PTB patients were diagnosed by CXR, 30% (78/263) by GeneXpert, and 7% (18/263) by smear. Moreover, five were confirmed by both smear and CXR, three by smear and GeneXpert, and 45 were confirmed by GeneXpert and CXR. Among the 263 PTB patients, 72% (190/263) were from Hospital A: 19% (37/190) were diagnosed by GeneXpert, 7% (13/190) by sputum smear and 72% (136/190) by CXR. Besides, 11% (22/190) were confirmed by both GeneXpert and CXR, 1% (3/190) by both GeneXpert and sputum smear, and 2% (4/190) by both sputum smear and CXR. Twenty eight percent (73/263) of the total PTB patients were from Hospital B: 52% (38/73) were diagnosed by GeneXpert, 3% (2/73) by sputum smear and 67% (49/73) by CXR. Moreover, 31% (23/73) were confirmed by both GeneXpert and CXR, 1% (1/73) was confirmed by sputum smear and CXR. At the study hospitals, TB microscopy was carried out in the onsite DOTs laboratory, and the TB drug susceptibility testing was performed at the chest disease hospital GeneXpert labs.

The median duration from hospital admission to final diagnosis was 4.6 days (IQR: 3.8 days), and to hospital discharge was 4.7 days (IQR: 3.8 days) (Fig 2) with a range from one to 22 days (Fig 3). In the univariate analysis, PTB patients' "length of hospital stay" significantly differed by hospital, age, and history of co-morbidities. In the multiple regression analysis, patients in Hospital A were two and a half times more likely to stay in the hospital greater than the median 4.7 days than patients in Hospital B (aOR -2.67, 95% CI:1.46–5.06). Patients aged

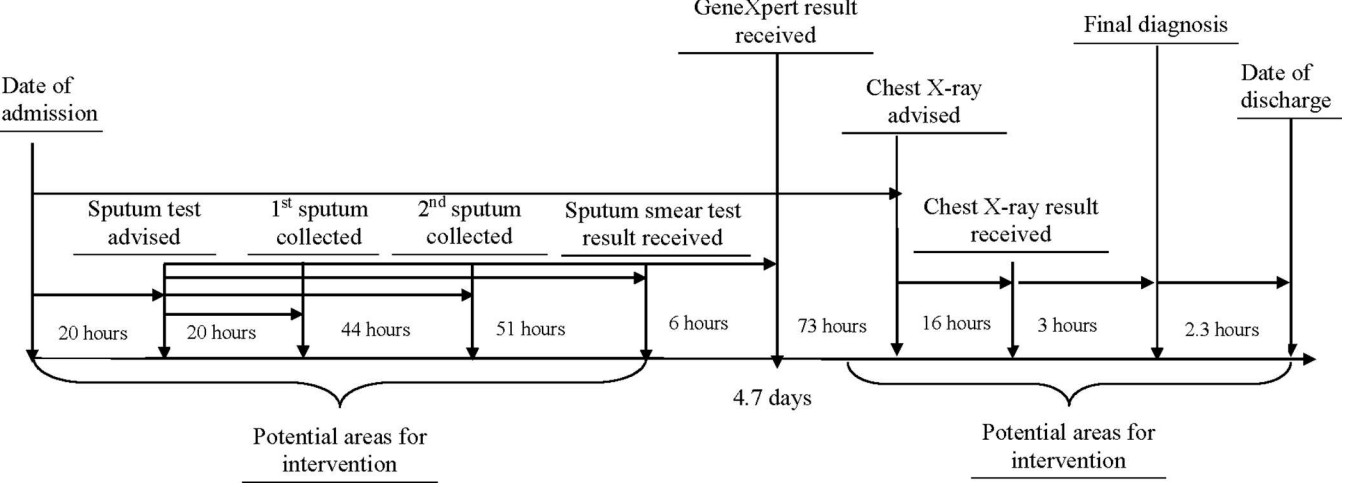

**Fig 2. Median time intervals between different activities from TB patient admission to discharge in two tertiary care hospitals, 2017–19.**

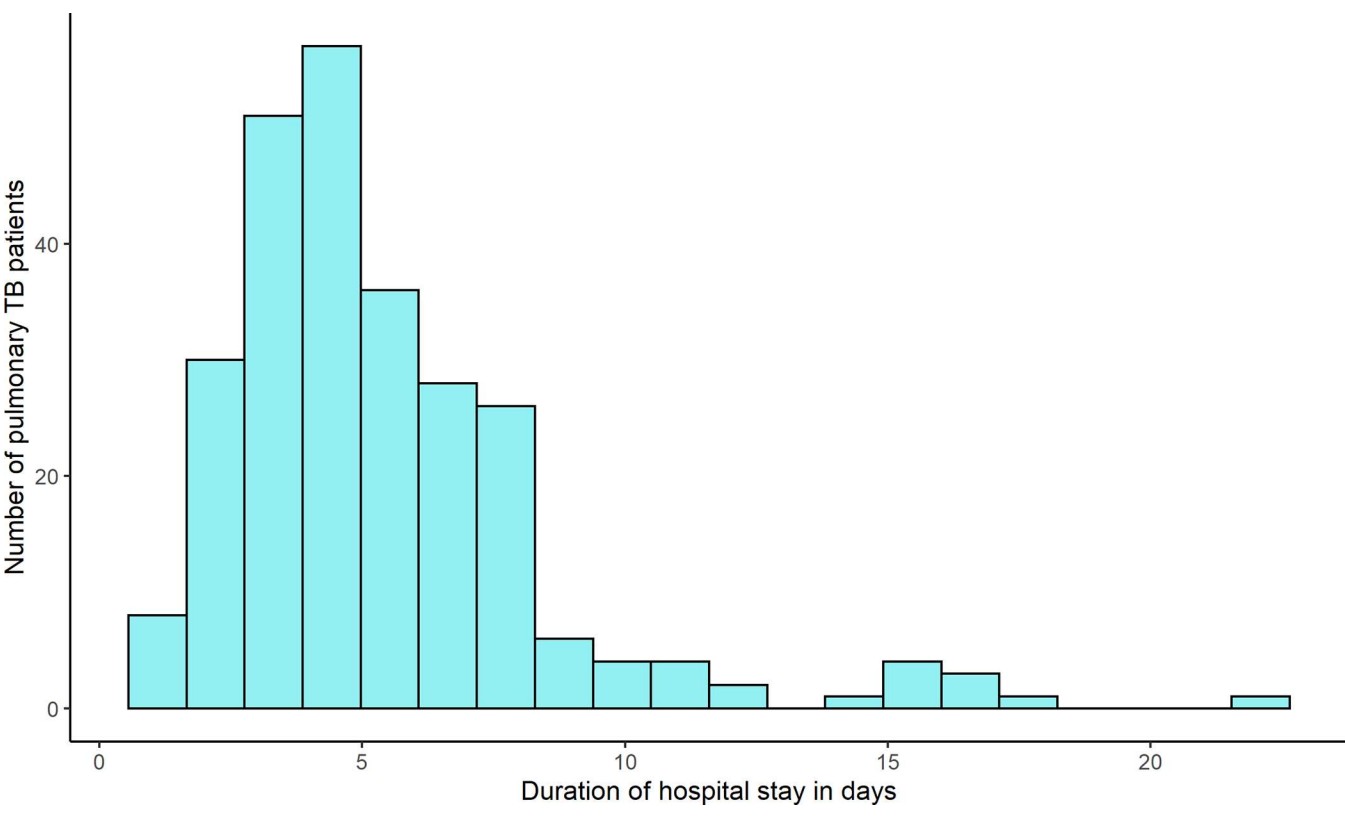

**Fig 3. Duration of pulmonary TB patient's hospital stay in two public tertiary care hospitals, Bangladesh.**

50–60 were also statistically significantly associated with a higher length of hospital stay (aOR3.06, 95% CI: 1.37–7.12). Moreover, patients with co-morbidities were 4.46 times more likely to stay greater than the median 4.7 days when compared with patients who did not report any comorbidity (aOR4.46, 95% CI:1.37–17.85) (Table 2).

## In-depth interview findings

Among 17 HCWs that participated in the in-depth interviews: ten HCWs from Hospital A and 7 were from Hospital B. The participants included hospital director, professors of medicine, emergency medical officers, nurse-in-charge, head of radiology departments, head of pathology department, ward registers and wards masters. The median age of the participants was 46 years (IQR:18), and the median years of experience in hospitals were 24 (IQR:17). The following is a thematic analysis of the findings from in-depth interviews.

**Presence of TB patients in general inpatient wards.**   Interview participants informed us that they occasionally treat TB patients in the study hospitals. One nurse explained the reasons for admitting and treating TB patients in general wards:

*"Occasionally, we receive some patients with TB and other co-morbidities or health complications. We receive TB patients with heart diseases. In such situations, we cannot refer patients to TB specialized hospitals. Instead, we treat them in our hospital."*

Another participant highlighted that on admission day, they receive six to eight patients with presumptive TB. Of these patients, only one or two are confirmed to have PTB disease.

**Table 2. Factors associated with pulmonary TB patients' length of hospital stay in two public tertiary care hospitals in Bangladesh, 2017–2019.**

| Demography and exposures | Patients with pulmonary TB (N = 263) %(n/N) | | OR (95% CI) | aOR (95% CI) | |
|---|---|---|---|---|---|
| | LOS< = 4.7 days | LOS>4.7 days | | | |
| **Hospitals** | | | | | |
| Hospital A | 45 (86/190) | 55 (104/190) | **2.02 (1.18–3.55)** | **2.67 (1.46–5.06)** | |
| Hospital B | 63 (45/72) | 37 (27/72) | Reference | | |
| **Gender** | | | | | |
| Male | 50 (58/117) | 50 (59/117) | Reference | | |
| Female | 50 (73/145) | 50 (72/145) | 0.97 (0.60–1.58) | | |
| **Age in years** | | | | | |
| <30 | 51 (40/78) | 49 (38/78) | Reference | | |
| 30–40 | 49 (19/39) | 51 (20/39) | 1.11 (0.61–1.48) | 1.03 (0.47–2.29) | |
| 40–50 | 66 (27/41) | 34 (14/41) | 0.55 (0.51–2.40) | 0.59 (0.26–1.30) | |
| 50–60 | 30 (14/46) | 70 (32/46) | **2.41 (1.13–5.30)** | **3.06 (1.37–7.12)** | |
| 60 and above | 53 (31/58) | 47 (27/58) | 0.92 (0.46–1.81) | 1.05 (0.52–2.12) | |
| **Comorbidity** | | | | | |
| No | 52 (127/246) | 48 (119/246) | Reference | | |
| Yes | 25 (4/16) | 75 (12/16) | **3.20 (1.08–11.70)** | **4.46 (1.37–17.85)** | |

LOS-Length of hospital stay, OR-odds ratio, aOR-adjusted odds ratio, CI-confidence interval.

All the participants mentioned that there were no isolation rooms for TB patients in the study hospitals. One nurse from Hospital A reported -

*"The TB IPC measures are not implemented properly in our hospital. We cannot isolate the TB patients at all due to limited resources and space. Therefore, the presumptive TB patients stay inside the ward with other patients. We want to isolate them, but we cannot manage it due to limited space."*

Others highlighted that TB patients walk around the ward and cough, sneeze, and spit without maintaining respiratory hygiene. They also added that some of the patients had multi-drug resistant TB.

**TB patients diagnostic delay.** The participants mentioned that it took around one week to diagnose a patient with TB. Two of the participants (both doctors) mentioned that patients are often admitted in the ward after office hours (i.e., after 2:30 pm) and do not receive a prescription for a TB test until the doctors' rounds on the following day. They added that several tests were required to confirm that a patient had TB disease. They said that a patient might receive a prescription for a sputum test, for TST, and X-ray on the same day but might not complete all the tests on the same day due to long waiting hours of the diagnostic centers. One of the participants described the process and the time required for TB diagnosis as-

*"We require morning and spot sputum sample to diagnose a patient with TB. We need 30 minutes to process the sample. It also requires one hour and 50 minutes to get GeneXpert and two to two and a half hours for acid-fast bacilli results. The patients receive the test results the next day."*

**Limited supply and poor use of masks and respirators.** Interview participants mentioned that there were no mask supplies available for patients. There was a supply of masks for

HCWs, but the participants reportedly did not use a mask regularly. One doctor explained why he did not use a mask in the ward-

*"I do not use any PPE during patients visit for patient satisfaction. If I use a mask, the patients feel discomfort. The patient may think about what disease he/she has for which I need to wear a mask."*

**High patient loads and family caregivers in the ward.**   The participants said that high patients load, and unrestricted visitors and family caregivers in the inpatient wards were the main challenges in IPC implementation in the hospital. One mentioned that the female medicine ward in Hospital A was at the highest risk of TB transmission due to high density in the ward. Another participant described the hospital crowd:

*"It is not possible to maintain a safe distance between patients or from patients to HCWs or family caregivers. On admission days, we had to allow patients on the floor, even in the corridor. During the ward round, the doctors cannot step in due to many patients on the floor. For example, this is a 30 bed-ward, and there should be 30patients, but 100 patients are staying in this 30-bed ward"*.

**TB among healthcare workers.**   All the interview participants informed us that HCWs were getting infected with TB disease due to their everyday exposures to PTB patients in the hospital. One participant said that she knew at least five nurses and nursing students who had been suffering from TB disease. The interview participants also mentioned that due to a lack of healthcare associated TB infection surveillance, the true number of TB disease among HCWs remained unknown in the hospital.

## Discussion

This study identified that public tertiary care general hospitals admit PTB patients on a regular basis. During hospital stay, most PTB patients remained untreated until diagnosis or discharge from the hospital. Untreated patients are most infectious, can contaminate confined spaces such as hospital inpatient wards and infect ward occupants including patients, HCWs and family caregivers [9, 26, 27]. Further, there was no use of masks among presumptive TB patients, which allowed the patients with pulmonary or laryngeal TB to contaminate the wards' air. Similarly, the reported TB disease among HCWs may be attributable to the exposure to untreated TB patients in the hospital wards. This theory is supported by the findings from a companion study conducted in the same hospitals, which found that 42% of the HCWs were positive with a tuberculin skin test and HCWs at the medicine wards were 3.65 (95% CCI: 2.01–6.79) times more likely to be positive with TST when compared with administrative workers in the hospital [28].

The high density observed in the inpatient wards along with a higher number of presumptive TB patients could facilitate the transmission of mycobacterium tuberculosis though uncovered coughing, sneezing and talking in the hospital [29]. A very limited use of respiratory protection among HCWs and patients was also observed, increasing the risk of TB exposures among ward occupants in the hospitals. The hospital record reviews revealed a higher number of PTB patients in Hospital B. Patients in Hospital B, patients aged between 50–60 years and patients with co-morbidities were statistically significantly associated with higher duration of hospital stay (>4.6 days). The findings from indepth interviews revealed the context for admitting PTB patients in these hospitals along with challenges these hospitals pose to follow recommended TB IPC measures. Thus, public tertiary care hospitals may facilitate the

transmission of TB among HCWs and other wards occupants and necessitate the immediate implementation of TB IPC measures. Hospital ward structure and crowding are two fundamental factors that may influence TB transmission [30]. The study wards were occupied by patients, family caregivers, children and HCWs, all of whom were coughing, sneezing, and spitting. A prior study found that the most crowded hospital locations were linked to a higher concentration of airborne bacteria [31]. International guidelines recommend the distance between patients bed should be at least 3.5 meters [32]. This recommendation is not easily followed in the study hospitals due to a high number of patients; instead, patients are allowed on the floor in between patients' beds. The proximity between patients and the mixing of PTB patients with patients without TB may increase the risk of TB transmission from patients to patients and from patients to family caregivers and children. Additionally, due to the continuous exposure to shared airspace, HCWs are at increased TB risk compared with other occupants.

Moreover, the number of family caregivers was twice the number of patients, making the inpatient ward more crowded. Low or no use of facemasks among family caregivers not only put them at risk of *Mycobacterium tuberculosis* exposures but also increases the possibilities of transferring the TB bacteria or other viruses to the ward occupants [33].

To minimize transmission to HCWs or other occupants in the ward, the WHO 2019 updated TB IPC guidelines recommend triaging presumptive TB patients to fast-track diagnosis and separate people with presumed infectious TB [9]. Both the hospitals lacked TB isolation rooms. The hospitals also lacked a consistent practice of separating patients with presumptive or confirmed infectious TB from non -infectious patients, increasing the risk of transmission from healthcare workers to non-TB patients [34]. In resource-limited settings where there is a lack of negative pressure isolation rooms, natural ventilation is preferred to prevent nosocomial transmission of TB [9, 35]. Although natural ventilation depends on outdoor weather conditions such as direction and magnitude of airflow, in some contexts, such as when mechanical ventilation systems are not appropriately maintained, it may be more effective at maintaining air flow [35, 36]. Our study found that the study wards had large doors and windows; however, most were partially or fully closed, preventing cross ventilation with fresh air. The minimum of 6 to 12 air change per hour required to control airborne infection control in the ward could not be ensured due to the closed doors and windows [35]. Even if the windows are closed on one side that prevents cross ventilation, large openings on the other side may not be inadequate to maintain the minimum air change [3].

This study identified that the median time from admission to final diagnosis for PTB patients was four and a half days, with 41% of PTB patients diagnosed after five days of admission. Several factors contributed to these diagnostic delays. First, the patients who arrived in the hospital after working hours could not manage a prescription for sputum or CXR and had to wait for a prescription until the next day. Second, both the study hospitals lacked 24-hour diagnostic services. Patients who reached the diagnostic facilities late could not do the test on the same day due to a long queue and waiting hours. Third, due to a lack of an electronic/online reporting system, hospital staff who are already overburdened with tasks in the ward, or the patients or their family caregivers, had to collect hard copies of the test report from the test centers. Finally, although most of the patients had a CXR suggestive of PTB, the HCWs often had to wait for sputum test results to finalize the TB diagnosis, which delayed the discharge or TB treatment. Due to all these health systems barriers, the PTB patients remained undiagnosed for nearly a week, and exhaled *Mycobacterium tuberculosis* through coughing and sneezing, which contaminated the ward environment as many patients did not cover their coughs and sneezes [37]. The TB bacteria may remain viable in the air for several hours and the ward occupants are likely to get infected through inhaling of the *Mycobacterium tuberculosis* [38].

The pulmonary TB patients' length of hospital stay significantly differed by hospitals which could be due their reliance on different TB diagnostic methods. The findings showed that Hospital A mostly relied on CXR and sputum smear results for diagnosis whereas Hospital B mostly relied on CXR and GeneXpert results which minimized the diagnostic delay. As shown in Fig 2, GeneXpert thus facilitated to reduce PTB patients' length of hospital stay, and this finding concurs with prior published literature [39]. The patients in the age group 50 to 60 stayed longer in the hospital when compared with the age group 30 and below and this could be because of older patients with PTB often do not present the typical signs and symptoms that delay diagnosis and increase the length of hospital stay [40]. PTB patient with co-morbidities is a known risk factor for increased length of hospital stay, and our findings further confirm this hypothesis [41].

This study has several limitations. First, this study was conducted in only two public tertiary care general hospitals. However, TB patient management, HCWs exposures and practices identified in these facilities are like chest diseases hospitals, other public and private facilities, and clinics in Bangladesh [1, 2, 42]. The findings from this study can be a strong indicator of what is happening in similar hospitals in Bangladesh and other low-income settings [42, 43]. Second, each field team members were responsible for observing 8–10 patients and their caregivers. The field team might likely have missed the frequency of coughs and sneezes that have co-occurred. Therefore, this study may have underestimated the actual number of coughing person and the frequency of coughs and sneezes. Finally, we evaluated the hospital level factors that contributed to diagnostic delay; patient-level factors may also contribute to hospital delay.

The ward occupants' long exposures to contaminated air space, possibly poor ventilation due to closed doors and windows, and lack of airborne IPC implementation in hospital wards all created an environment that increased the risk of TB transmission. This study warrants the immediate implementation of TB IPC measures in public tertiary care hospitals, starting with adult medicine wards. As a first step, administrative control measures should be prioritized, including screening patients with chronic cough, triage of presumptive TB patients, fast-track diagnostic service for presumptive TB patients, keeping doors, windows, and other openings open 24 hours a day in all seasons. Ancillary workers can be utilized to ensure proper natural ventilation in the hospital wards. We also recommend hospitals establish airborne isolation rooms for TB and other airborne infectious diseases. If an isolation room is not possible, hospitals should separate presumptive TB patients either in the corner of the ward near windows with adequate natural ventilation or a veranda with an open window to the outside. An electronic reporting and recording system for laboratory test result sharing can be implemented in these facilities to improve diagnostic delay. In each ward, a few nurses can be designated as 'cough officer' who can screen patients with a history of cough more than two weeks for presumptive TB and may recommend sputum evaluation in consultation with a duty doctor. Similar interventions have been successful in identifying presumptive TB patients quickly in general hospitals [44]. Our study also found that smear-based diagnosis increased the diagnostic delay as it required multiple samples. We thus recommend testing all presumptive TB patients with GeneXpert. We acknowledge that GeneXpert is expensive (USD 50 per test); however, the use of GeneXpert will reduce the length of hospital stay from several days to hours and, thus, will be more cost-effective. We also recommend a patient with PTB should be discharged within an hour from final diagnosis, or PTB patients should be isolated immediately after diagnosis confirmation. Finally, PPE and basic education and training on respiratory hygiene and cough etiquette should be provided to both HCWs and patients with respiratory symptoms.

## Acknowledgments

We would like to thank the study hospitals' directors and all the study participants for their time and respect. icddr,b acknowledges with gratitude the commitment of CDC to its research efforts. icddr,b is also grateful to the Governments of Bangladesh, Canada, Sweden, and the UK for providing core/unrestricted support.

## Author Contributions

**Conceptualization:** Md. Saiful Islam, Sayera Banu, Abrar Ahmad Chughtai, Holly Seale.

**Data curation:** Md. Saiful Islam, Kamal Ibne Amin Chowdhury, Arifa Nazneen, Mohammad Tauhidul Islam, S. M. Zafor Shafique, S. M. Hasibul Islam.

**Formal analysis:** Md. Saiful Islam, Sayera Banu, Sayeeda Tarannum, Abrar Ahmad Chughtai, Holly Seale.

**Funding acquisition:** Md. Saiful Islam, Sayera Banu.

**Investigation:** Mohammad Tauhidul Islam.

**Methodology:** Md. Saiful Islam, Kamal Ibne Amin Chowdhury, Mohammad Tauhidul Islam, Abrar Ahmad Chughtai, Holly Seale.

**Project administration:** Md. Saiful Islam, Sayeeda Tarannum, Kamal Ibne Amin Chowdhury, Arifa Nazneen, Mohammad Tauhidul Islam, S. M. Zafor Shafique, S. M. Hasibul Islam, Holly Seale.

**Supervision:** Md. Saiful Islam, Sayera Banu, Abrar Ahmad Chughtai, Holly Seale.

**Validation:** Sayeeda Tarannum, Kamal Ibne Amin Chowdhury, Arifa Nazneen, S. M. Zafor Shafique, S. M. Hasibul Islam, Abrar Ahmad Chughtai, Holly Seale.

**Visualization:** Md. Saiful Islam.

**Writing – original draft:** Md. Saiful Islam.

**Writing – review & editing:** Md. Saiful Islam, Sayera Banu, Sayeeda Tarannum, Kamal Ibne Amin Chowdhury, Arifa Nazneen, Mohammad Tauhidul Islam, S. M. Zafor Shafique, S. M. Hasibul Islam, Abrar Ahmad Chughtai, Holly Seale.

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
