## [Decision Letter · Decision Letter 0]

1 Oct 2021

*PGPH-D-21-00420*

*Examining pulmonary TB patient management and healthcare workers exposures in two public tertiary care hospitals, Bangladesh*

*PLOS Global Public Health*

**Dear Dr. Md Saiful Islam,**

Thank you for submitting your manuscript to PLOS Global Public Health. After careful consideration, we feel that it has merit but does not fully meet PLOS Global Public Health’s publication criteria as it currently stands. Therefore, we invite you to submit a revised version of the manuscript that addresses the points raised during the review process.

Please **submit your revised manuscript by November 13th, 2021**. If you will need more time than this to complete your revisions, please reply to this message or contact the journal office at globalpubhealth@plos.org. Please include these changes in your article, replying to all of the comments made, before a final decision is made concerning the paper's acceptance or refusal.

The revised version will be reevaluated by the original reviewers.

We look forward to receiving your revised manuscript.

Kind regards,

Raquel Muñiz-Salazar, Ph.D.

Academic Editor

Journal Requirements:

1. Thank you for including your ethics statement "We sought informed written consent before the interviews. The institutional review board of the International Centre for Diarrheal Diseases Research, Bangladesh (icddr,b), and the

University of New South Wales ethical review committee approved the study (PR-16090 and HC

# HC180517)."

In your ethics statement, please also state whether written informed consent was obtained from the patients whose medical records were observed, or participants included in the observational sessions. If the ethics committee waived the need for informed consent in these cases, please state this.

2. Please provide separate figure files in .tif or .eps format only.

3. Since your data is not available for proprietary reasons, please explain via email why the data is not available. Please also include the contact information for the third party organization that should be contacted should other researchers want to request access to this data and please include the full citation of where the data can be found. We also request that you verify with us via email that any researcher will be able to obtain the data set in the same manner that the you have obtained it. If you feel you are unwilling or unable to adhere to this policy, please explain your reasons by return email and your exemption request will be escalated to the editor for approval. Your exemption request will be handled independently and will not hold up the peer review process, but will need to be resolved should your manuscript be accepted for publication. One of the Editorial team will be in touch if they require more information.

4. Please amend your detailed Financial Disclosure statement. This is published with the article, therefore should be completed in full sentences and contain the exact wording you wish to be published.

i) State the initials, alongside each funding source, of each author to receive each grant.

Reviewers' comments:

**Comments to the Author**

1. Does this manuscript meet PLOS Global Public Health’s publication criteria? Is the manuscript technically sound, and do the data support the conclusions? The manuscript must describe methodologically and ethically rigorous research with conclusions that are appropriately drawn based on the data presented.

Reviewer #1: Yes

Reviewer #2: Yes

2. Has the statistical analysis been performed appropriately and rigorously?

Reviewer #1: Yes

Reviewer #2: Yes

3. Have the authors made all data underlying the findings in their manuscript fully available (please refer to the Data Availability Statement at the start of the manuscript PDF file)?

Reviewer #1: Yes

Reviewer #2: Yes

4. Is the manuscript presented in an intelligible fashion and written in standard English?

Reviewer #1: No

Reviewer #2: Yes

**5. Review Comments to the Author**

**Reviewer #1: **In their manuscript, titled “Examining pulmonary TB patient management and healthcare workers exposures in two public tertiary care hospitals, Bangladesh”, Islam and colleagues have provided a wide range of evidence for implementing tuberculosis (TB) infection prevention and control (IPC) in public tertiary care general hospitals in Bangladesh. Their study uses several methods, including direct observation, a detailed review of patient records, and in depth interviews with hospital staff in two study sites in order to gain an understanding of current TB management. Their results reveal that current practice is indeed insufficient to curb TB transmission, especially in inpatient wards. The study’s utility in informing IPC guidelines is clear; the authors conclude by recommending a comprehensive set of actions to minimize TB transmission, which may be generalized to other healthcare settings in Bangladesh and other resource-limited contexts. Following some methodological clarification and additional analysis, which I elaborate on below, this study will provide a clear case for TB IPC implementation in public tertiary care hospitals in Bangladesh.

Minor Comments:

There are several instances in each section of the manuscript where I suggest adjusting basic wording and punctuation in order to meet PLoS guidelines for clarity; each of these instances is listed below.

Abstract:

• In the first line of the introduction, please replace “TB” with “tuberculosis (TB)”.

• In the last line of the introduction, please add replace “hospitals, Bangladesh” with “hospitals in Bangladesh”.

• In the results section, please replace “contaminate” with “contaminated” in the first sentence.

• In the results section, please end the sentence after the word “mask”. Then, the following sentence should be, “Further, patients did not wear any respirators”.

• In the conclusion, consider adding “for some duration of time” to the end of the first sentence. Saying that most TB patients stayed in the hospitals untreated implies that patients always remained untreated throughout their entire stay.

Introduction:

• In Line 6, please replace “drug-resistance” with “drug-resistant”.

• In Line 10, consider removing the sentence that starts with “Diagnostic delay”, as the same thought is conveyed two sentences later.

• In Line 12, please start the sentence with “A lack of screening procedures”.

• In Line 14, please add an apostrophe after wards: “hospital wards’ occupants”.

• In Line 23 (top of Page 5), please replace “co-morbid TB patients” with “TB patients with comorbidities”.

• In Lin 25, please add an apostrophe after healthcare workers so that it reads “healthcare workers’ (HCW)”.

• In Line 26, please add “Bangladesh” so that it reads “in Bangladesh’s public tertiary care”.

• In Line 27, please add an apostrophe after patients so that it read “TB patients’ stay”

• In Line 28, please reword to “TB patient’s length of hospital stay and management, healthcare workers’ exposures to”.

Methods:

• In Line 36 (near the top of Page 6), please change “experiences” to “experience”.

• In Line 41, please replace “diagnostic” with “diagnosis”.

• In Line 42, at the end of the sentence, consider adding “within these facilities”, so that it reads “(there was already an established relationship and research occurring within these facilities)”.

• In Line 44 (in the Observations section), how many adult general medicine wards did the team conduct observations in? I suggest adding in brackets “(n male wards and n female wards)”.

• In line 57 (in the Hospital record review section), please remove “the” from “We reviewed the hospital records”.

• In Line 59, please replace “registers” with “registries”.

• In Line 61, please replace “CXR” with “chest x-ray (CXR)”, as I believe this is the first time that the readers see this abbreviation.

• In Line 62, please replace “the” with “those” so that it reads “diagnosed with TB among those suspected”.

• In Line 64, please add “testing” so that it reads “and the percentage testing positive”.

• In Line 65, please add “the” so that is reads “the team talked to the patient’s family”.

• In Line 68 (in the In-depth interviews section), please replace “purposively” with “purposefully”

• In Line 76, please replace “them from tuberculosis” with “TB transmission”.

• In Line 81 (in the Analysis section), please replace “spend” with “spent”.

Results:

• In Line 100, please state whether the numbers pertain to male wards, female wards, or both.

• Please be consistent with bracket use – currently there are both square and circular brackets being used, i.e. [] and ().

• In Line 101, please add an IQR for the 10 HCWs.

• In Lines 102-103, please consider rewording as follows “A median of 3.35 (IQR: 2.87) occupants were present per 10 m2 of floor space, with the highest number of occupants observed in female medicine wards in Hospital A; 7.0 (IQR: 0.7) occupants per 10 m2 of floor space”.

• In Line 106, please replace “was” and “and”, and change “them” to “whom”, so that it reads “The team recorded a median of 15 people (IQR: 15) coughing during the observation period, most of whom were patients”.

• In Line 109, please superscript the “2” in “10 m2”.

• In Line 109, please add “of” so that it reads “Only 8.4% of”.

• In Line 118, please add “having” so that it reads “identified as having presumptive TB”.

• In Line 118, please replace “and” with “, of which” so that it reads “over 20 months, of which 22%”.

• In Line 120, please add “were” so that it reads “55% were male”.

• In Lines 128-129, is there a reason why mean hospitalization duration was reported instead of median hospitalization duration?

• In Line 131, please add “that” so that it reads “Among 17 HCWs that participated”.

• In Line 134, please reword as “The average age of the participants was 46.1 years”. Please also add a standard deviation to this. Is there a reason why mean was reported instead of median?

• Following Line 134, consider adding a sentence to this effect: “What follows is a thematic analysis of the in-depth interviews.”

• In Line 141, please change “as” to “with” so that it reads “six to eight patients with presumptive TB”.

• In Line 142, consider rewording as such: “Of these patients, only one or two are confirmed to have PTB disease”.

• In Line 153 (in the TB patients diagnostic delay section), please replace “a” with “one” so that it reads “around one week”.

• In Lines 154-157, consider rewording as such: “Two of the participants (both doctors) mentioned that patients are often admitted in the ward after office hours (i.e. after 2:30 pm) and do not receive a prescription for a TB test until the doctors’ rounds on the following day. They added that several tests were required to confirm that a patient had TB disease.”

Discussion:

• In Line 187, consider rewording as such: “This study identified that public tertiary care general hospitals admit PTB patients on a regular basis”.

• In Line 190, please replace “ward” with “wards”.

• In Line 191, please add “Further” and “which”, so that it reads “Further, there was no use of masks among presumptive B patients, which allowed”.

• In Line 194, please replace “supporting” with “supported”.

• In Line 208, please replace “Besides” with “Additionally”.

• Line 218 (“natural ventilation could be more effective than mechanical ventilation if not appropriately maintained”) is unclear. Do you mean that, if mechanical ventilation is not appropriately maintained, natural ventilation is more effective at maintaining air flow?

• In Line 222, please reword to: “…could not be ensured due to the closed windows and doors”.

• In Line 223, please replace “close” with “closed”.

• In Line 229, please add “and” so that it reads “chest X-ray and had to wait”.

• In Line 235, please add “a” so that it reads “had a chest X-ray”.

• In Line 236, please replace “that delay” with “which delayed”.

• In Line 238, please add “and”, “which” and “ward” so that it reads: “…remained undiagnosed for nearly a week, and exhaled mycobacterium tuberculosis through coughing and sneezing, which contaminated the ward environment”. Consider adding a sentence after this that mentions that many patients did not cover their coughs or sneezes.

• In Line 244, please replace “clinic” with “clinics”.

• In Line 245, please replace “indicative” with “indicator”.

• In Line 250, please replace “cough” with “coughs”.

• In Line 253, please replace “close” with “closed”.

• In Line 261, please add “an” so that it reads “If an isolation room”.

• In Line 263, please add “a” so that it reads “or a veranda”.

• In Line 268, please replace “similar intervention had” with “similar interventions have”.

General:

• On the title page, you have written “Key-wards” instead of “Key Words”.

• Please be consistent with your use of “inpatient” or “in-patient”.

• Please be consistent with number formatting. In general, if the number is less than 10, you can write it out, e.g. “two” or “three”. If it’s at the beginning of a sentence, it can be written out as well, even if it’s larger than 10. In all other instances, it can be referred to numeric format, e.g. “15” or “25”.

• Please capitalize “iccdr,b”.

• Since you’ve already specified early in the paper that chest x-ray is abbreviated as “CXR”, any time it is mentioned afterwards, it can be referred to as CXR instead of chest x-ray.

• When referring to the hospitals, please use “Hospital A” and “Hospital B” instead of “hospital A” and “hospital B”.

Tables and Figures:

• Please add standard deviation to average number of HCWs per ward in Table 1.

• Do the numbers in square brackets in the “Average number of windows closed” row represent standard deviation? If so, please add a footnote to this effect in Table 1.

• The x-axis of Figure 1 should read “Observation dates” instead of “Observations dates”.

Major Comments:

• The three-pronged approach to answering your research question is interesting, holistic, and engaging to the reader. However, a reader that is not familiar with mixed methods may not understand the rationale behind doing observations, chart reviews, and in-depth interviews. I suggest adding a brief section – it could be separate or integrated with your “Study design, study site, and data collection” section – that explains why each of the three methods was chosen, and what specific objectives each of the three methods would meet.

• The authors do a great job of summarizing their quantitative data using descriptive statistics. However, there are some questions that remain unanswered, which would help strengthen their conclusions. For example, are there important differences between Hospital A and B that limit their comparability? Are there important differences between characteristics of male and female wards? Are there factors that are statistically associated with a higher number of TB patients per ward? Are there factors statistically associated with delayed diagnosis? These are just examples of questions that would help strengthen the authors’ recommendations at the end if answered. I suggest adding some additional analysis to accompany the quantitative results (e.g. regression).

• In the discussion, the authors talk about generalizing conclusions to the broader healthcare context in Bangladesh and potentially other resource-limited settings. How representative are Hospital A and Hospital B compared to other tertiary hospitals in Bangladesh? I suggest adding a sentence or two to this effect. A good place to put this information would be the end of your Methods section’s first paragraph (where the authors provide rationale for selecting these hospitals).

• At (or near) the beginning of the discussion, I suggest briefly listing the main takeaway messages from each of the three approaches (observations, hospital record review and in-depth interviews). This will help the reader tie everything together before you proceed to discuss further detail.

• In the “Hospital record review” section of the results, you refer to Figure 2, which is a great depiction of timelines. The authors show in the picture that there are some areas for potential intervention, but don’t really touch on this in the text. I suggest adding a sentence or two to mention what kinds of interventions could be implemented at different time points, with reference to Figure 2.

**Reviewer #2:** The study appears to be sound, a statistical description of the quality of care for TB patients, reporting a structured observation that identifies the critical points of intervention, such as reinforcing education for the proper use of protective equipment, both for the health personnel as well as on the part of the patients, and the little application of hygienic measures by the patients.The results show the obvious need to monitor hospital overcrowding and the proper operation of naturally ventilated wards.

The statistical analysis and process is adequate, however, it is necessary to review the figures and their relevance in the conclusions. For example, figure 1 is only cited once in the results and does not impact the discussion, in addition, the information is not clear, this figure indicates on the x-axis that the observations were in a period of two months from December 15 to 15 February, but the study lasted 2 years (December 2017 to July 2019). I recommend reviewing this aspect and possibly considering substituting for the thematic map used in the study.

*While revising your submission, please upload your figure files to the Preflight Analysis and Conversion Engine (PACE) digital diagnostic tool, https://pacev2.apexcovantage.com/. PACE helps ensure that figures meet PLOS requirements. To use PACE, you must first register as a user. Registration is free. Then, login and navigate to the UPLOAD tab, where you will find detailed instructions on how to use the tool. If you encounter any issues or have any questions when using PACE, please email PLOS at figures@plos.org. Please note that Supporting Information files do not need this step.*

---

## [Decision Letter · Decision Letter 1]

23 Nov 2021

Examining pulmonary TB patient management and healthcare workers exposures in two public tertiary care hospitals, Bangladesh

PGPH-D-21-00420R1

Dear Dr. Islam,

We're pleased to inform you that your manuscript has been judged scientifically suitable for publication and will be formally accepted for publication once it meets all outstanding technical requirements.

Within one week, you'll receive an e-mail detailing the required amendments. When these have been addressed, you'll receive a formal acceptance letter and your manuscript will be scheduled for publication.

An invoice for payment will follow shortly after the formal acceptance. To ensure an efficient process, please log into Editorial Manager at https://www.editorialmanager.com/pgph/ click the 'Update My Information' link at the top of the page, and double check that your user information is up-to-date. If you have any billing related questions, please contact our Author Billing department directly at authorbilling@plos.org.

Kind regards,

Raquel Muñiz-Salazar, Ph.D.

Academic Editor

Additional Editor Comments (optional):

Dear Md Saiful Islam

I am pleased to inform you that your manuscript "Examining pulmonary TB patient management and healthcare workers exposures i two public tertiary care hospitals, Bangladesh" has been accepted for publication in PLOS Global Public Health.

Reviewers' comments:

Reviewer's Responses to Questions

**Comments to the Author**

1. If the authors have adequately addressed your comments raised in a previous round of review and you feel that this manuscript is now acceptable for publication, you may indicate that here to bypass the “Comments to the Author” section, enter your conflict of interest statement in the “Confidential to Editor” section, and submit your "Accept" recommendation.

Reviewer #1: All comments have been addressed

Reviewer #2: All comments have been addressed

2. Does this manuscript meet PLOS Global Public Health’s publication criteria? Is the manuscript technically sound, and do the data support the conclusions? The manuscript must describe methodologically and ethically rigorous research with conclusions that are appropriately drawn based on the data presented.

Reviewer #1: Yes

Reviewer #2: Yes

3. Has the statistical analysis been performed appropriately and rigorously?

Reviewer #1: Yes

Reviewer #2: Yes

4. Have the authors made all data underlying the findings in their manuscript fully available (please refer to the Data Availability Statement at the start of the manuscript PDF file)?

Reviewer #1: Yes

Reviewer #2: Yes

5. Is the manuscript presented in an intelligible fashion and written in standard English?

Reviewer #1: Yes

Reviewer #2: Yes

6. Review Comments to the Author

Reviewer #1: The authors have adequately addressed all comments. Congratulations on completing this important study!

Reviewer #2: Comments were addressed appropriately

7. PLOS authors have the option to publish the peer review history of their article (what does this mean?). If published, this will include your full peer review and any attached files.

**Do you want your identity to be public for this peer review?** For information about this choice, including consent withdrawal, please see our Privacy Policy.

Reviewer #1: **Yes: **Aashna Uppal

Reviewer #2: No
